The seeker R package: simplified fetching and processing of transcriptome data

http://orcid.org/0000-0001-8544-6359 Schoenbachler Joshua L. 1
http://orcid.org/0000-0002-1558-6089 Hughey Jacob J. 1 2 jakejhughey@gmail.com
1 Department of Biomedical Informatics, Vanderbilt University Medical Center , Nashville, Tennessee , United States
2 Department of Biological Sciences, Vanderbilt University , Nashville, Tennessee , United States
Sun Hong-Wei
Electronic publication date: 2022 Nov 7
Publication date: 2022
Volume: 10
Electronic Location ID: e14372
Received 2022 Sep 1; Accepted 2022 Oct 19
Copyright: © 2022 Schoenbachler and Hughey
Copyright year: 2022
Copyright holder: Schoenbachler and Hughey
License: This is an open access article distributed under the terms of the Creative Commons Attribution License, which permits unrestricted use, distribution, reproduction and adaptation in any medium and for any purpose provided that it is properly attributed. For attribution, the original author(s), title, publication source (PeerJ) and either DOI or URL of the article must be cited.
License URL: https://creativecommons.org/licenses/by/4.0/

Keywords: Transcriptome data, Automation, Command-line interface, Genomics, RNA-seq, Microarray

Funding: National Institute of General Medical Sciences R35GM124685 This work was supported by the National Institute of General Medical Sciences (R35GM124685). The funders had no role in study design, data collection and analysis, decision to publish, or preparation of the manuscript.

==============================
Transcriptome data have become invaluable for interrogating biological systems. Preparing a transcriptome dataset for analysis, particularly an RNA-seq dataset, entails multiple steps and software programs, each with its own command-line interface (CLI). Although these CLIs are powerful, they often require shell scripting for automation and parallelization, which can have a high learning curve, especially when the details of the CLIs vary from one tool to another. However, many individuals working with transcriptome data are already familiar with R due to the plethora and popularity of R-based tools for analyzing biological data. Thus, we developed an R package called seeker for simplified fetching and processing of RNA-seq and microarray data. Seeker is a wrapper around various existing tools, and provides a standard interface, simple parallelization, and detailed logging. Seeker’s primary output—sample metadata and gene expression values based on Entrez or Ensembl Gene IDs—can be directly plugged into a differential expression analysis. To maximize reproducibility, seeker is available as a standalone R package and in a Docker image that includes all dependencies, both of which are accessible at https://seeker.hugheylab.org.

Introduction

Measuring the transcriptome has become a standard approach for understanding biological systems. However, even after a transcriptome dataset is generated, it must go through multiple processing steps before it is ready for analysis. For RNA-seq data in particular, each step typically involves a different software program with its own command-line interface (CLI). Thus, complete processing of transcriptome data requires chaining together the output of one program with the input of another. Automating and parallelizing the entire process has often involved shell scripting, which can have a steep learning curve, limiting reproducibility and portability. To address this issue and simplify RNA-seq processing pipelines, two programs—pyrpipe (Singh et al., 2021) and nf-core/rnaseq (Ewels et al., 2020)—have recently been developed that wrap around various RNA-seq processing programs. pyrpipe uses Python, whereas nf-core/rnaseq uses Nextflow. However, many individuals working with transcriptome data are already familiar with R, given the popularity of R-based tools for biological data analysis, especially for quantifying differential expression. Thus, we developed an R package called seeker, which simplifies the fetching and processing of RNA-seq and microarray data. Seeker provides an R-based interface for various existing programs and enables simple automation and parallelization.

Methods

The seeker package’s website (https://seeker.hugheylab.org) includes instructions for installing the package, multiple vignettes, and detailed documentation for every function.

System dependencies

For convenience, the seeker package includes a function called installSystemDeps that can install and configure the package’s system dependencies: the NCBI SRA Toolkit (which includes the tools prefetch and fasterq-dump; https://github.com/ncbi/sra-tools), Miniconda (with Python 3; https://docs.conda.io/en/latest/miniconda.html), the Mamba package manager (https://github.com/mamba-org/mamba), and the conda packages fastq-screen (Wingett), fastqc (Andrews), multiqc (Ewels et al., 2016), pigz, refgenie (Stolarczyk et al., 2020), salmon (Patro et al., 2017), and trim-galore (Krueger). installSysDeps sets environmental variables so the CLIs are accessible within R, and installs snakemake (Mölder et al., 2021) to make it easier to use seeker in reproducible analyses. We also provide a Docker image called socker, based on rocker/rstudio from the Rocker Project, in which seeker and its dependencies are already installed.

The seeker package has no hardware requirements other than those of its system dependencies and of R itself. In general, processing RNA-seq data is more memory- and compute-intensive than processing microarray data. However, the memory requirements and computation time to process a given RNA-seq dataset will depend on one’s hardware, the number of files to process and their size, the extent of parallelization, and which processing steps are being run. For specific estimates, we refer readers to the papers and documentation of the underlying tools.

Microarray data

To fetch and process microarray data, the package includes a function called seekerArray (Fig. 1A), which depends on the GEOquery and ArrayExpress R packages. seekerArray can process data from NCBI GEO or Array Express, or raw Affymetrix data stored locally. The main inputs to seekerArray are a study accession and a preferred gene ID type (Entrez or Ensembl). The main outputs are a table of sample metadata and a matrix of log2-transformed gene expression measurements. seekerArray automatically detects the microarray platform and maps probes to the appropriate gene IDs. If the study includes raw Affymetrix data, seekerArray performs RMA (Irizarry, 2003) and uses custom CDFs from Brainarray (Dai, 2005), otherwise seekerArray uses the study’s processed data.

Figure 1 Schematics of the seekerArray and seeker functions of the seeker R package.

(A) The preferred option requires raw Affymetrix data. The fallback option uses processed data. (B) The main inputs and outputs are listed for each function.

RNA-seq data

To fetch and process RNA-seq data, the package includes multiple functions (described below), each of which handles single-end or paired end reads, includes sensible defaults, and allows the user to pass custom arguments to the respective APIs or CLIs. fetchMetadata: Uses the API of the European Nucleotide Archive (ENA) or the Sequence Read Archive (SRA) to fetch a study’s sample metadata. The main input is a Bioproject accession.

fetch: Uses srafetch, fasterq-dump, and pigz to download files from SRA and convert them to gzipped fastq files in parallel. The main input is a vector of SRA run accessions.

trimgalore: Uses Trim Galore to perform standard adapter and quality trimming in parallel. The main input is a vector of paths to fastq files.

fastqc: Uses FastQC to perform quality control checks in parallel. The main input is a vector of paths to fastq files.

salmon: Uses Salmon to quantify transcript abundances in parallel. The main inputs are a vector of paths to fastq files, corresponding sample names, and a directory containing the salmon transcriptome index (which can be fetched by refgenie via installSysDeps).

multiqc: Uses MultiQC to aggregate the results of various processing steps (including Trim Galore, FastQC, and Salmon) into a single report. The main input is a directory containing the results.

getTx2gene: Uses the biomaRt package (Durinck et al., 2009) to create a mapping of transcripts to genes based on Ensembl IDs. The main input is an organism name.

tximport: Uses the tximport R package (Soneson, Love & Robinson, 2015) to summarize Salmon’s transcript-level quantifications for gene-level analyses. The main inputs are a directory containing quantification directories from Salmon and a mapping of transcripts to genes.

The most computationally intensive functions (fetch, trimgalore, fastqc, and salmon) output their progress as tab-delimited text files.

In addition, the package includes a function called seeker, which can perform any of the above steps and pass the output of one step as input to the next step (Fig. 1B). The main input to the seeker function is a list of parameters, which can be derived from a yaml file, that specifies which steps to perform and how to perform them. Depending on the steps specified, the seeker function can fetch and process publicly available RNA-seq data or process locally stored data. The function also includes a “dry run” option to check the parameters’ validity without fetching or processing any data.

Reproducibility

To help ensure that the output of seekerArray and seeker is reproducible, both functions save a yaml file of the user-defined parameters and a text file containing the R session information. The latter is provided by the sessioninfo R package and includes version numbers and sources of all loaded packages. In addition, the seeker function saves a text file containing paths and version numbers of its system dependencies, as well as a yaml file of the conda environment, which includes version numbers of the miniconda-based dependencies.

To provide an even higher level of reproducibility, the seeker package and its dependencies are available in a Docker image called socker (https://github.com/hugheylab/socker), which is based on rocker/rstudio (https://hub.docker.com/r/rocker/rstudio). Using socker thus ensures that the output is not dependent on details of the local computing environment (Nüst et al., 2020).

Results

To illustrate the utility of the seeker package, we used it to fetch and process multiple publicly available transcriptome datasets. We first used the seekerArray function to fetch two datasets related to circadian metabolism in mouse liver (GSE34018 and GSE67964) (Cho et al., 2012; Zhang et al., 2015). The main output for GSE34018, which is based on an Illumina beadchip, consisted of 24 samples and 17,142 Entrez genes. The main output for GSE67964, which is based on an Affymetrix array, consisted of eight samples and 27,352 Entrez genes. We used these outputs to perform principal components analysis and to calculate differential expression using limma (Ritchie et al., 2015) between the wild-type and knockout samples in each dataset (Fig. 2). The log2 fold-changes for Entrez genes measured in the two datasets were weakly negatively correlated (Spearman’s rho −0.092), consistent with the opposing roles of the genes knocked out in each dataset (Nr1d1 and Nr1d2 in GSE34018; Rora and Rorc in GSE67964).

Figure 2 Example of using the output of seekerArray.

Principal components analysis of (A) GSE34018 and (B) GSE67964, both of which are microarray datasets based on livers of mice in a 12 h:12 h light:dark cycle. Each point represents a sample. In (A), color indicates time since lights on and shape indicates genotype (the knockout was liver-specific). In (B), color and shape indicate genotype (all samples were collected at 22 h after lights on). (C) Scatterplot of log2 fold-changes of differential expression between wild-type and knockout for the 16,747 Entrez genes measured in both datasets. Each point represents a gene.

We next used the seeker function to fetch and process one sample from each of two RNA-seq datasets related to circadian rhythms and feeding in mouse liver (PRJNA600892 and PRJNA667743) (Guan et al., 2020; Manella et al., 2021). We chose two samples that had the same genotype (wild-type), feeding regimen (ad libitum), and time of day of acquisition (4 h after lights on). PRJNA600892 is based on paired-end sequencing and Illumina TruSeq Stranded library prep, whereas PRJNA667743 is based on single-end sequencing and bulk MARS-Seq (3′-tagged) library prep (Jaitin et al., 2014). The seeker function detected and handled the paired-end and single-end reads automatically. To account for the 3′-tagged reads in PRJNA667743, we specified the countsFromAbundance argument to tximport as “no” in the input parameters to seeker. For simplicity and speed, we disabled the trimgalore step and used a salmon index based on partial selective alignment. The main outputs were the sample metadata and the tximport object including counts and abundances for 35,494 Ensembl genes. As expected, the gene-level abundances between the samples from the two datasets were highly correlated (Fig. 3; Spearman’s rho 0.877).

Figure 3 Example of using the output of seeker.

Scatterplot of abundance for 35,494 Ensembl genes in sample SAMN13836337 (from PRJNA600892) and sample SAMN16382424 (from PRJNA667743). Each point represents a gene. TPM stands for transcripts per million.

Discussion

The seeker package complements existing software packages for fetching and processing transcriptome data. Whereas pyrpipe and nf-core/rnaseq are focused on RNA-seq data, seeker can also process microarray data. Here seeker builds on the GEOquery and ArrayExpress R packages and our previous package metapredict (Hughey & Butte, 2015) by providing a simple interface to map microarray probe sets to standard gene IDs. In addition, unlike pyrpipe, seeker can run in parallel natively, without requiring a workflow manager such as Nextflow. seeker also allows the user to specify all processing parameters for a given dataset in a single yaml file. Perhaps most importantly, seeker uses the R language and computing environment, and thus could appeal to a wider group of researchers already using R for bioinformatic analyses.

The seeker package does have limitations. First, the current implementation focuses on processing bulk transcriptome data using a relatively small set of tools. Although these should suffice for the majority of use cases, seeker currently supports a more limited set than either pyrpipe or nf-core/rnaseq. For example, it supports only salmon for quantifying transcript-level abundances, which is a standard in the field and is both fast and accurate. In the future, we plan to extend seeker to accommodate other types of genomic data, including from single cells, and a wider range of tools. Second, although seeker thoroughly documents its computing environment, recreating that environment on another machine (if not using the socker Docker image) would still be a manual process. We welcome contributions from the community to improve seeker in these and any other ways. By providing a straightforward, R-based approach to small- or large-scale processing of transcriptome data, we hope seeker contributes to improved reproducibility and transparency of genomic analyses.

Reproducible results are available at https://doi.org/10.6084/m9.figshare.20720848.

Additional Information and Declarations

Competing Interests

Author Contributions

Data Availability

Jacob J. Hughey is an Academic Editor for PeerJ.

Joshua L. Schoenbachler performed the experiments, analyzed the data, authored or reviewed drafts of the article, and approved the final draft.

Jacob J. Hughey conceived and designed the experiments, performed the experiments, analyzed the data, prepared figures and/or tables, authored or reviewed drafts of the article, and approved the final draft.

The following information was supplied regarding data availability:

The reproducible results are available at Figshare: Hughey, Jacob (2022): Reproducible results for: The seeker R package: simplified fetching and processing of transcriptome data. figshare. Dataset. https://doi.org/10.6084/m9.figshare.20720848.v2.

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
