# Peer review of "The seeker R package: simplified fetching and processing of transcriptome data"

_PeerJ, doi:10.7717/peerj.14372_

## Round 0.1 · original submission · Major Revisions

The reviewers recognize the general and practical utility of the seeker R package and think it could be a useful tool to the relevant research community for early-stage RNA-seq and microarray data analysis. It is also believed that for a wrapper tool, particular attention should be given to make it user friendly and to avoid adding another layer of complexity. This is especially important given that most of the expected users of this R package would be tool users working in a lab, rather than experienced bioinformaticians.

The reviewers have constructively suggested revisions to improve the manuscript and to make the R package better documented and more user friendly. They have also raised several issues encountered during their testing of the software package. Please provide detailed responses, point by point, to reviewers’ suggestions and the issues they have raised.

·

Basic reporting

In this manuscript, the authors have presented a useful R-based tool for handling RNA-seq and microarray data. The data could be de-nova or downloaded from a public repository like GEO. If users’ interest is to download, preprocess and analyze the existing data from GEO, the fetching, quality check, mapping, counting the gene expression, etc. are made easy using this package. One can fully automatize the process by providing all parameters through yaml file and implementing seeker or seekerArray functions. Alternatively, one can process the data step by step using separate functions. The functions are written to log most of the computation steps and their progress status. The analysis is reproducible.

The abstract and introduction sections are well written. Formats of citation, table and figures are good.

Experimental design

Looks good.

Validity of the findings

Looks good. Since authors are not testing noble hypothesis using de-novo experimental data, the content of this section might be less important.

Additional comments

There are several elements lacking in this manuscript. The points of improvement are discussed below.

Tutorial document
a. It is hard to navigate different steps like installing the packages and dependencies and implementing the functions. For example, the author should demonstrate step by step procedures like:

· hardware requirements
· install the package using xxx commands from git hub
· install dependencies using installSysDeps
· check if all cmds is available in the computing environment using checkDefaultCommands

A separate tutorial document with workflow is highly recommended to make navigation easy even for novice users. The authors should mention the availability of the tutorials in this manuscript.

Result section
b. The manuscript has listed some steps that require more computation time; however, computation time has not been discussed for seeker and seekerArray implementation. The computation time depends on file size and the number of files being processed, but the authors should provide general computation time and memory requirements for the commonly used file size.

R script
c. fetchMetadata implementation throws standard output but does not create table or csv file.
d. ?function_name does not show the example.

Discussion section
e. The authors did not discuss why particular packages or software were used to build this tool but not others. For example, the use cases of SALMON and STAR could be different. Why salmon is pipelined in this package not STAR? What is the pros and cons of FASTQC over MULITIQC?
f. There are several tools available to analyze RNA-seq data. The authors should critically analyze and present the pros and cons of this package over some other popular packages.

Overall impression
This manuscript can be accepted for publication only after a major revision and expansion of the sections to incorporate the above suggestions. The minor changes do not make this manuscript publishable in peerj. This package is very useful, and I suggest expanding the manuscript and asking for additional documentation to make this package more comprehensive for all.

Reviewer 2 ·

Basic reporting

The manuscript presents an R package for downloading and carrying out some basic analyses of microarray and RNAseq data from NCBI and European Nucleotide Archive (ENA). The writing style of the manuscript is good and the code depository is well established. However, the figures are not informative, and codes are not tested thoroughly. I suggest the author have a major revision for the manuscript. Detail comments are as follows:

Experimental design

1) No other R packages performing the similar RNAseq analysis were mentioned in the introduction. Those comparisons are needed to highlight the unique contribution of Package Seeker.

2) Microarray pipeline works well. The microarray data were obtained from NCBI. RNAseq pipeline was not run successfully in both Docker and local HPC environments. No data were downloaded. More testing and better instruction are needed.

Validity of the findings

3) Both figures are not relevant to the pipeline. The simple PCA and scatter plots make the manuscript look weak. A pipeline workflow will be more useful. Other figures to demonstrate the utility will be helpful too.

4) There are no examples in the reference document. More and better examples in vignette and reference document will be needed for understanding and testing the package. BiocManager documentation style is a good example.

---

## Round 0.2 · accepted · Accept

The reviewers are satisfied with the revised version that has clearly improved as they have constructively suggested. It has also successfully addressed all the reviewers’ concerns.

It is my belief that the revised manuscript meets the publication criteria of PeerJ and it is now ready to be published in the journal.

·

Basic reporting

No comments

Experimental design

No comments

Validity of the findings

No comments

Additional comments

The authors have made changes to the important comments.

Reviewer 2 ·

Basic reporting

The revision has improved the software instruction.

Experimental design

The figures are improved and more informative now.

Validity of the findings

The software bug was fixed and throughly tested.

Additional comments

The revision has addressed the issues and improved the instruction and quality of the R package Seeker.